# Evaluation of the Metabolite Profile of Fish Oil Omega-3 Fatty Acids (n-3 FAs) in Micellar and Enteric-Coated Forms—A Randomized, Cross-Over Human Study

**DOI:** 10.3390/metabo14050265

**Published:** 2024-05-07

**Authors:** Afoke Ibi, Chuck Chang, Yun Chai Kuo, Yiming Zhang, Min Du, Yoon Seok Roh, Roland Gahler, Mary Hardy, Julia Solnier

**Affiliations:** 1ISURA, Clinical Research, Burnaby, BC V3N 4S9, Canada; aibi@isura.ca (A.I.);; 2Factors Group R & D, Burnaby, BC V3N 4S9, Canada; 3Academy of Integrative and Holistic Medicine, San Diego, CA 92037, USA

**Keywords:** bioavailability, eicosapentaenoic acid, docosahexaenoic acid, fish oil, LipoMicel, micellar, n-3 fatty acids, n-3 metabolites

## Abstract

This study evaluated the differences in the metabolite profile of three n-3 FA fish oil formulations in 12 healthy participants: (1) standard softgels (STD) providing 600 mg n-3 FA; (2) enteric-coated softgels (ENT) providing 600 mg n-3 FA; (3) a new micellar formulation (LMF) providing 374 mg n-3 FA. The pharmacokinetics (PKs), such as the area under the plot of plasma concentration (AUC), and the peak blood concentration (C_max_) of the different FA metabolites including HDHAs, HETEs, HEPEs, RvD1, RvD5, RvE1, and RvE2, were determined over a total period of 24 h. Blood concentrations of EPA (26,920.0 ± 10,021.0 ng/mL·h) were significantly higher with respect to AUC_0-24_ following LMF treatment vs STD and ENT; when measured incrementally, blood concentrations of total n-3 FAs (EPA/DHA/DPA3) up to 11 times higher were observed for LMF vs STD (iAUC _0-24_: 16,150.0 ± 5454.0 vs 1498.9 ± 443.0; *p* ≤ 0.0001). Significant differences in n-3 metabolites including oxylipins were found between STD and LMF with respect to 12-HEPE, 9-HEPE, 12-HETE, and RvD1; 9-HEPE levels were significantly higher following the STD vs. ENT treatment. Furthermore, within the scope of this study, changes in blood lipid levels (i.e., cholesterol, triglycerides, LDL, and HDL) were monitored in participants for up to 120 h post-treatment; a significant decrease in serum triglycerides was detected in participants (~20%) following the LMF treatment; no significant deviations from the baseline were detected for all the other lipid biomarkers in any of the treatment groups. Despite a lower administered dose, LMF provided higher blood concentrations of n-3 FAs and certain anti-inflammatory n-3 metabolites in human participants—potentially leading to better health outcomes.

## 1. Introduction

Omega-3 fatty acids (n-3 FA) represent essential polyunsaturated fatty acids (PUFAs) that have anti-inflammatory properties, with many reported health benefits for humans [1,2]. The highest dietary sources of n-3 FAs—specifically eicosapentaenoic acid (EPA) and docosahexaenoic acid (DHA), which are associated with numerous health benefits e.g., cardiovascular and cognitive improvements—are typically found in marine sources like fatty fish, algal oil, or krill oil [3,4,5,6]. Alpha-linolenic acid (ALA) is more commonly found in plant-based sources like chia seeds, flaxseeds, and walnuts, and needs to be converted by the body into a usable form (i.e., EPA and DHA), with a rather poor conversion rate of 5–15% [7,8]. Docosapentaenoic acid (DPA3 or n-3 DPA) is found in fish oil, but at much lower concentrations compared to EPA and DHA. Though DPA3 has been less extensively studied, emerging research suggests that it provides similar health benefits to EPA and DHA [9].

Recent studies have also drawn focus to the clinical effects of n-3 FA metabolites including oxylipins, a diverse group of bioactive lipid mediators derived from the oxidation of polyunsaturated fatty acids (PUFAs). Some metabolites of DHA include hydroxy-docosahexaenoic acids (HDHAs), Resolvin D1 (RvD1) and Resolvin D5 (RvD5), all of which have shown anti-inflammatory effects in in vitro studies [10]. Resolvins such as RvD1 and RvD5 play a role in pain management by reducing hypersensitivity and modulating pain signals [11,12], and having cardiovascular protective effects as well as neuroprotective properties—which may be particularly important in connection to the development of autoimmune diseases [13,14]. Similarly, Resolvin E1 (RvE1) and Resolvin E2 (RvE2) derived from hydroxyeicosapentaenoic acids (HEPEs)—metabolites of EPA—are known for their anti-inflammatory and immune modulating properties [15,16]. Other oxylipins including HETEs (hydroxyeicosatetraenoic acids), classified as hydroxyeicosanoids, are part of the larger family of eicosanoids, synthesized from arachidonic acid (AA). HETEs play a role in the pro-inflammatory signaling pathways contributing to the initiation and propagation of the inflammatory response. In vitro studies have reported that HETEs serve as a chemoattractant, attracting immune cells to the site of inflammation, and have been implicated in processes related to cell proliferation and differentiation, as well as platelet activation [17]. This can lead to pathological conditions such as thrombosis, as well as diseases associated with oxidative stress such as Parkinson’s disease and Alzheimer’s disease [18,19]. Furthermore, HETEs have been associated with inducing obesity, coronary artery disease, and breast cancer [20,21,22]. 

Modern Western diets are usually low in n-3 FAs, especially EPA and DHA, and high in saturated fats, trans fats and Omega-6 polyunsaturated fats [23]. A deficiency in essential n-3 FAs can lead to an array of health problems related to cardiovascular health, cognitive development, inflammatory conditions, autoimmune diseases and cancer [24,25]. Further compounding this deficiency are those with certain pre-existing health conditions or those who live solely on a plant-based diet, making supplementation desirable [26]. 

The most commonly studied n-3 FA forms are naturally occurring and reconstituted triglycerides (TG), ethyl esters (EE), and free fatty acids (FFA). Pharmacokinetic studies reported that the most bioavailable form of n-3 FA is when it is provided in the order of FFA > TG > EE [27,28,29]. 

The absorption of n-3 FAs in supplement form can be influenced by several factors. Put simplistically, following the oral administration of n-3 FAs in humans, bile acids aid emulsification and pancreatic enzymes break down the larger triacylglycerol fat molecules into smaller free fatty acids that can be more easily absorbed through the small intestinal epithelium. The efficiency of this process can vary between individuals due to physiological and genetic differences [30,31]. Another consideration is the formulation used for oral supplementation. For example, n-3 FAs, especially in the free fatty acid form, have been shown to be more prone to peroxidation, which can lead to stability issues, undesirable byproducts, as well as reduced bioavailability [32,33]. N-3 supplements formulated in delayed release forms may help to create a gastro-resistant barrier and reduce fishy aftertaste or smell [34]. Enteric-coated (ENT) n-3 FAs are resistant to stomach acid and appear to offer increased absorption [35,36]. The enteric coating of softgel capsules involves covering the capsules in a coating of synthetic polymers or natural ingredients, which prevent premature rupture and the subsequent degradation of active compounds in the acidic environment of the stomach to allow for its release in the more neutral or alkaline environment of the small intestine [37]. Some studies have shown that the otherwise common use of enteric-coated softgel capsules for fish oils appears to have little to no impact on bioavailability. For instance, Schneider et al., using enteric-coated capsules of EPA- and DHA-rich fish oils, did not find significant differences in bioavailability in comparison to standard capsules [38]. More recently, however, there has been evidence that emulsified delivery matrices containing micelles of either TGs and EEs are able to provide superior oral bioavailability over regular fish oils alone [39,40]. 

The aim of this study is to determine the pharmacokinetic (PK) differences in n-3 FAs and metabolites of three different fish oil formulations: Omega-3 triglycerides in standard soft-gelatin capsules (i.e., the most common market form); Omega-3 triglycerides in enteric-coated soft-gelatin capsules (i.e., an improved formulation to the standard form to reduce aftertaste and smell); and Omega-3 triglycerides in a novel micellar matrix soft-gelatin capsule (namely LipoMicel^®^). The primary outcome measures include the PKs such as AUC0-24 and C_max_ of the n-3 FAs including EPA, DHA, as well as their metabolites such as HDHAs, HEPEs, RvD1, RvD5, RvE1, and RvE2, determined in human participants. Additionally, changes in blood lipid parameters (e.g., cholesterol and related lipids) were monitored up to 120 h (5 days) post-dose. 

## 2. Materials and Methods

### 2.1. Omega-3 (n-3) Formulations

Three commercially available fish oil supplements containing Omega-3 triglycerides were examined for this study. Omega-3 Complete softgel capsules (non-enteric, standard (STD)) were purchased from Jamieson Wellness Inc. (Toronto, ON, Canada); Extra Strength RxOmega-3 softgel capsules (enteric-coated (ENT)) were purchased from Natural Factors (Burnaby, BC, Canada); Omega-3 micellar (LipoMicel^®^ (LMF)) softgel capsules were newly formulated and provided by Natural Factors (Burnaby, BC, Canada).

Standard (STD) softgel capsules are composed of 1000 mg fish oil (molecularly distilled; anchovy, tuna), containing 600 mg Omega-3 fatty acids which provide: 400 mg of EPA and 200 mg of DHA; other ingredients are gelatin (bovine) and glycerin.Enteric-coated (ENT) softgel capsules are composed of 1170 mg fish oil (molecularly distilled, ultra-purified; anchovy, sardine, and/or mackerel), containing 600 mg Omega-3 fatty acids which provide: 400 mg EPA and 200 mg DHA; other ingredients are gelatin, glycerin, purified water, pectin, and natural vitamin E (patented Enteripure^®^ softgels).Micellar (LMF) softgel capsules are composed of 585 mg fish oil (molecularly distilled), containing 374 mg of Omega-3 fatty acids which provide: 200 mg EPA, 133 mg DHA and 41 mg DPA3; other ingredients are glycerin, water, gelatin bovine bone, cocoa powder, xylitol, medium chain triglycerides, and methylsulfonylmethane (patent pending LipoMicel^®^).

The single dose used in the study was one capsule of n-3 fish oil/day (max. 600 mg). The administered amount of n-3 FAs varied among treatments—with LMF containing a total lower amount of n-3 FA with a ratio of EPA:DHA 2:1.5 compared to the 2:1 in other treatments.

### 2.2. Study Design

This study is a randomized, double blind, crossover study (Figure 1). Table 1 provides the inclusion and exclusion criteria for this study. All participants provided a written consent form and completed an online health questionnaire on their medical history prior to participation. The study was approved by the Canadian SHIELD Ethics Review Board (OHRP Registration IORG0003491; FDA Registration IRB00004157; Approval letter ID 2021-10-002, date of approval: 28 February 2022). The study has been registered on ClinicalTrials.gov with Identifier NCT05394701 and conducted in accordance with the ethical standards as set forth in the Helsinki Declaration of 1975.

In Phase I of the study, the pharmacokinetics of different n-3 FA supplements were evaluated by collecting capillary whole blood samples over a period of 24 h: at baseline (t = 0 h), 0.5, 1, 2, 3, 4, 6, 8, 12 and 24 h post-dose (Figure 2). In Phase II, participants continued taking the treatments daily for 5 consecutive days or up to 120 h to monitor changes in blood lipids. Capillary blood samples were collected at baseline (t = 0 h), 3, 6, 12, 24, 48, 72, 96, and 120 h post-dose (Figure 2). A two-week washout period was employed before each subsequent n-3 treatment.

A single dose of each n-3 supplement was administered to each participant with at least 9 h of overnight fasting. Participants were blinded and did not know the size and shape of each individual treatment beforehand, so that they were incapable of identifying the treatments. Treatments were consumed in the morning with a glass of water, and within 30 min were followed by a standardized breakfast, consisting of a bagel with cream cheese and jam. Standardized lunch was provided for the first 24 h of the treatments; standardized breakfast and dinner were provided throughout the entire study period as summarized in Table 2. 

### 2.3. Determination of Omega-3 Fatty Acids and Metabolites in Blood

The samples were prepared and analyzed according to previously published methods [22,41,42]. Whole blood samples (50 µL) were hydrolyzed with 300 µL 10 M sodium hydroxide (Sigma, Oakville, ON, Canada) at 60 °C for 30 min. Then, samples were neutralized with 300 µL 60% acetic acid (Sigma, Canada), and 50 µL of diluted internal standard solution was added prior to hydrophilic–lipophilic balance (HLB)-type Solid Phase Extraction (SPE, Phenomenex, Torrance, CA, USA). Samples applied to SPE were washed with water, 15% methanol, and hexane prior to elution with ethyl formate. The eluted fatty acid-rich extract was then evaporated under nitrogen and reconstituted with 50 µL of ethanol.

The reconstituted samples were then injected at 20.0 µL volumes into a Thermo Vanquish Ultra High-Performance Liquid Chromatography system (UHPLC, Thermo Scientific, Montreal, QC, Canada) from microplates kept at 10.0 °C and separated on 100 × 2.1 mm Acme Xceed C18, 1.9 µm particle columns (Phase Analytical Technology, State College, PA, USA) that are temperature controlled in a 40.0 °C column oven. A binary mobile phase, consisting of 0.5% formic acid in water (Mobile Phase A) and acetonitrile (Mobile Phase B), was used with a linear gradient from 50–70% Mobile Phase B over the first 4.00 min, and then 70–80% from 4.00 to 5.00 min, 80–95% from 5.00 to 8.00 min, and with an isocratic gradient of 95% B from 8.00 to 10.00 min. All reagents used were LCMS (Liquid Chromatography Mass Spectrometry) grade and obtained from Fisher Scientific, Canada. N-3 Fatty acids, metabolite chemical standards and deuterated internal standards were purchased from Cayman Chemical (Ann Arbor, MI, USA). See Table 3 for the list of chemical standards used.

The UHPLC was connected to a Thermo Q exactive Orbitrap mass spectrometer set at 35,000 mass resolution in Parallel Reaction Monitoring (PRM) mode using an isolation window of 0.4 *m*/*z* and a normalized collision energy of 27 eV. Detailed scan parameters are described in Table 3. A heated electrospray ion source (HESI) was operated in negative mode, and calibration was carried out with Thermo Scientific™ Pierce™ LTQ Velos ESI Negative Ion Calibration Solution (Thermo Scientific, Canada) weekly with a typical mass error less than 1 ppm. Appendix A contains the mass extraction parameters for the complete list of n-3 FA and their metabolites. Sample chromatograms for these compounds can be found in Appendix A.

### 2.4. Determination of Blood Lipids

Blood lipids were determined using a Cholestech LDX Analyzer with Cholestech LDX Lipid Profile Cassettes (Abbott, Princeton, NJ, USA). Briefly, 40 µL of capillary blood was drawn into a heparinized glass capillary (Alere, Waltham, MA, USA), and injected into single-use analysis cassettes. The cassette was placed into the analyzer which then provided results within about 5 min. The analyzer was calibrated according to the manufacturer’s instructions prior to use each day.

### 2.5. Data Analysis

The main pharmacokinetic parameter examined was the area under the plot of plasma concentration of the formulation versus 24 h after dosage (AUC_0-24_). The mean value for AUC_0-24_ of each individual Omega-3 active and metabolite was calculated and expressed as graphs. Blood concentration AUCs of n-3 FAs and metabolites were evaluated for normality using the Kolmogorov–Smirnov test with an alpha of 0.05. Non-normal AUCs were log transformed prior to statistical analysis with a repeated measure two-way ANOVA followed by a post hoc Tukey’s multiple comparison test. Changes in blood lipids compared to baseline values were analyzed with a repeated measure two-way ANOVA followed by a post hoc Dunnett’s multiple comparison correction.

### 2.6. Randomization and Blinding

Randomization was performed using Research Randomizer from https://www.randomizer.org/. Sixteen sets of numbers were generated, and each set contained a series of 3 non-repeating numbers from 1 to 3. Each number represented three different treatments: Standard (STD) soft gel capsules, enteric-coated (ENT) soft gel capsules, and micellar (LMF) soft gel capsules. Treatments were dispensed to the participants according to the sequence of the number series in each treatment week.

## 3. Results

### 3.1. Baseline Characteristics

16 participants were initially recruited and randomized to treatment (9 men; 7 women; average age 36 years with mean BMI of 22.1 ± 0.5 kg/m^2^ and mean weight of 61.8 ± 2.4 kg; Table 4); 12 participants completed the phase I trial and were included in the pharmacokinetic analysis. Due to the much longer time commitment required, in phase II, only a small number of participants (*n* = 8) completed the blood lipid analysis as demonstrated in Figure 2.

### 3.2. Pharmacokinetics of n-3 FAs

The average blood concentrations over time for the n-3 FAs such as DHA, EPA, and DPA3 following the oral administration of each treatment were analyzed and expressed as bar graphs (Figure 3). Of notable interest was the AUC_0-24_ for LMF, which had the highest total blood concentration of total actives and was up to 4-fold higher compared to the other treatments (Table 5). While the DHA and DPA3 concentrations do not differ significantly among the treatment groups, the EPA blood concentrations are up to ~ 5 times higher in the LMF group compared with those of the ENT group and ~ 6 times higher than those of the STD group (Figure 3 and Table 5).

When measured incrementally, with baseline subtracted, LMF achieved up to 8- and 11-fold higher blood concentrations of total n-3 FAs, i.e., iAUC of EPA/DHA/DPA3, compared to ENT and STD, respectively (Figure 4 and Table 6). Notably, as opposed to the other treatments, LMF achieved peak concentrations at 10 h (T_max_; Table 6). Both STD and ENT treatments displayed more gradual and stable blood concentrations (Figure 4) with peak levels at ~4 h and 5 h, respectively (T_max_; Table 6). In comparison, LMF illustrated an early peak after 2 h, suggesting a more rapid absorption, as well as later peaks at 6 and 8 h (Figure 4), likely resulting from the lunch at 4 h, which may suggest a release from intestinal epithelial cells where n-3 FAs may have been stored [30].

Furthermore, LMF still showed higher blood concentrations at 24 h compared to the non-micellar formulation, STD and ENT (Figure 4).

### 3.3. Blood Concentrations of Metabolites

Blood concentrations of 17 individual metabolites were analyzed over the first 3 h and 24 h (Figure 5 and Figure 6). Significant differences were found between STD and LMF over the first 3 h with respect to 12-HEPE, 12-HETE, and RvD1; 9-HEPE levels were significantly higher following the STD vs. ENT treatment (Table 7).

While the blood concentrations of most of the metabolites were consistently found to be relatively low after 24 h, significant differences were only found with the mean AUC_0-24_ of RvD1, with the highest concentrations of this metabolite being observed after STD treatment (Table 8). The most considerable difference was found between LMF and the other treatments where there were significantly lower concentrations of RvD1.

Although not significant, STD demonstrated the highest average AUC_0-24_ in terms of the total metabolites: HDHA, HETE, and HEPE, as well as HHTrHe.

### 3.4. Blood Lipids

The blood lipids were analyzed with respect to HDL cholesterol, LDL cholesterol, LDL/HDL ratio, total cholesterol (TC), triglycerides (TRG), and non-HDL cholesterol. Figure 7 compares the aggregate participant results for each treatment after five days. After five consecutive days (120 h) of LMF treatment, participants showed a significant reduction in serum triglycerides (−19.5%; *p* ≤ 0.01) when compared against their individual Time 0 values. The other blood lipid parameters did not show any significant changes (Figure 7). STD and ENT did not produce any significant changes in the blood parameters of participants after 5 days.

Appendix A present the detailed serum lipid parameters results on a day-by-day basis. Significant changes from baseline values were observed mostly in TRG values where LMF treatment at 48, 72, 96, and 120 h showed consistent significant decreases. The changes for the other two treatments were more transient, with STD treatment showing a one-time significant decrease at 48 h, and ENT treatment at 12 h.

HDL, LDL, LDL/HDL ratio, TC, and non-HDL cholesterol did not show any significant changes from the baseline when the data were analyzed based on each treatment (Appendix A). However, when the data from all three treatments were aggregated together, HDL showed significant decreases at 6, 12, 24, and 120 h. LDL showed a significant decrease at 12 h. The LDL/HDL ratio showed a significant increase at 120 h, TC showed a significant decrease at 96 h, and TRG showed a significant increase at 12 h.

### 3.5. Side Effects

A safety survey was performed to assess any adverse events which occurred during the study. No side effects were reported during the study.

## 4. Discussion

The primary goal of this study was to establish pharmacokinetic parameters and analyze the metabolite profiles of different Omega-3 formulations in human participants over a 24 hr study period. Results showed that a micellar formulation (LipoMicel, LMF) of microencapsulated n-3 FAs achieved the highest total concentrations of DHA, DPA3 and EPA following the oral administration of fish oil softgel capsules. Compared to standard (STD) and enteric-coated (ENT) formulations administered at a higher dose of n-3 FAs (i.e., 600 mg vs 374 mg), LMF achieved significantly higher AUC_0-24_ of EPA (approx. five to seven times higher than the other treatments), and when measured incrementally, up to 11 times higher blood concentrations of total n-3 FAs (EPA/DHA/DPA3).

These results are similar to the approximately three-fold improvement in absorption found in another randomized trial on DHA and EPA formulated in a self-emulsifying delivery system called PhytoMarineCelle (PM). At a dosage of 500 mg, the AUC_0-24_ for PM was 106.62 ± 1.92 µg·h /mL for EPA, 79.53 ± 2.32 µg·h /mL for DHA and 186 ± 1.67 µg·h /mL for EPA+DHA [43]. Another study investigating the effects of administering 4.5 g microencapsulated EPA-rich fish oils over 24 h reported an approximated 10 times increase in AUC_0-24_ compared to standard fish oil capsules, from 2 ± 1.4 mg·h /100 mL to 19.7 ± 4.3 mg·h /100 mL, proposing that the finely dispersed oil droplets of the microencapsulated fish oil are easier to break down by lipase [44]. Given that the dispersion of fat globules is a prerequisite for the digestion of lipids, the ingestion of a lipid microemulsion such as that created by LMF serves to simplify this physiological process, allowing for better digestion and absorption of n-3 FAs [32]. The significantly larger difference in the AUC_0-24_ for EPA in LMF as compared to the other formulations may be enough to consider it a major rich source of supplementary EPA—with potentially greater health benefits for humans. Studies have revealed that 26 weeks of continued supplementation with EPA-rich oil can lead to improved overall cognitive function in terms of speed and accuracy, especially in the case of healthy young adults [45].

While the AUC_0-24_ of DHA was almost 1.5 times higher in LMF than the other formulations, although not significant, its overall concentration across all three formulations was significantly lower than its counterpart, EPA. Studies have shown that DHA is more susceptible to oxidation than EPA, and this may in effect contribute to its reduced absorption in the gastrointestinal tract. However, those same studies also highlighted that the ratio of EPA to DHA may also play a factor in the stability of the FA mixture. A 1:1 ratio of EPA:DHA is more stable in the stomach because it results in less conjugated dienes. Consequently, a more balanced ratio of the two n-3 FAs may lead to a higher resistance to oxidation [46]. Hence, the 2:1.5 ratio of EPA:DHA in LMF compared to the 2:1 ratio in the other two formulations may be a contributing factor to its better performance.

LMF, which provides a microemulsion of n-3 FAs, was demonstrated to have the highest bioavailability among the formulations tested in this study. One possible explanation may be that microemulsions, characterized as colloidal systems comprising oil and water phases stabilized by surfactants, enhance the solubility and dispersibility of lipophilic substances such as Omega-3 fatty acids in the gastrointestinal tract [47]. Furthermore, the smaller droplet size and uniform dispersion of Omega-3 fatty acids within the oil phase of microemulsions may amplify the available surface area for interactions with digestive enzymes, thereby promoting both the rate and extent of absorption [48]. Enteric coatings of formulations like ENT, on the other hand, are designed to delay the release of contents until they reach the small intestine, but the dispersion of the oil within the capsule may not be as uniform, potentially affecting absorption.

EPA and DHA are considered generally safe for long-term consumption at combined doses of up to about 5 g/day [49,50]. With LMF providing a lower dose of EPA and DHA per capsule but showing higher bioavailability, a corresponding lower therapeutic dose or different dosing regimen may be used compared to standard treatment.

Interestingly, as observed in this study, even after 24 h post-dose, LMF still showed significantly greater blood concentrations of n-3 FAs (Figure 4). The prolonged presence of n-3 FAs in the bloodstream 24 h after administration with a microemulsion formulation suggests a preferential release of n-3 FAs such as EPA from the liver back into the bloodstream. This is a common phenomenon that has been observed in other studies as well as mentioned by Aarak et al. [51]. Due to the lipophilic nature of n-3 fatty acids, they may integrate into lipid-rich compartments within cells or tissues. Therefore, one of the limitations of this study is that a more extended observation period would have been essential to attain a more comprehensive understanding of the metabolic trajectory of n-3 fatty acids as formulated in LMF.

In terms of the metabolite profiles, although LMF had the highest bioavailability based on significantly higher AUCs of total n-3 FAs, i.e., EPA, DHA and DPA3, it did not show significant blood concentrations of the total EPA and DHA-derived metabolites after 24 h. This was likely due to a delayed metabolism of LMF (microencapsulated fish oil n-3 FA) as illustrated in Figure 4. Accordingly, a longer study period (>24 h) should be applied in follow-up studies. Chuang et al. observed a similar seemingly delayed fatty acid metabolism in their 24 h study on a lipid-based delivery system [43]. Therefore, in this study, only a few significant differences in the metabolite concentrations were observed between LMF and the other treatments—most of these were observed within the first 3 h. For instance, LMF showed significantly lower concentrations of 12-HETE compared to STD. Lower concentrations of 12-HETE, a specific oxylipin derived from the oxidation of arachidonic acid, can be seen as a positive effect, as studies have linked higher concentrations of 12-HETE to cancer cell proliferation and the metastasis of tumor cells [52,53,54].

The only significant increase following LMF treatment compared to STD was noticed in 12-HEPE (approx. five times greater AUC_0-3_). 12-HEPE may be a key n-3 metabolite in glucose metabolism, as certain studies have shown that production of this metabolite is repressed in obese subjects [55]. Shaikh et al. highlighted studies which demonstrated that increase in 12-HEPE can improve the uptake of glucose into skeletal muscle and brown adipose tissue, consequently leading to improved glucose tolerance, an important benefit for diabetics [56]. However, future studies are yet needed to further understand the overall role 12-HEPE plays in glucose regulation and its potential benefits to those with glucose intolerance.

Additionally, in this study, changes in blood lipids (e.g., cholesterol, LDL and HDL) were monitored in participants up to 120 h (or five days) post-treatment. No significant changes were observed from the baseline, except for those observed in the serum triglycerides (~20% reduction), following the LMF treatment. A previous study by Balk et al. examining the effects of fish oil consumption in 21 different trials ranging from 7 to 104 weeks found that consistent intake of n-3 FAs produced an average reduction in serum triglycerides by about 27 mg/dL. Furthermore, they observed that this decrease in triglycerides was concurrently linked to a modest increase in LDL (6 mg/dL) and HDL (1.6 mg/dL) cholesterol; however, the increase in HDL and LDL levels are moderate enough that they do not pose a sufficient independent risk to cardiovascular health. The study also showed that n-3 FAs had no significant effect on total cholesterol [57]. In our study we observed a similar significant decrease in serum triglycerides (0.32 mmol/L). However, given that this was only a pilot investigation into the effects of n-3 FAs on blood lipids with a very small or limited sample size, a future investigation would benefit from having an increased sample size and a longer observation period.

This research concludes that, of the three formulations analyzed in this study, LMF demonstrated the highest absorption of total n-3 FAs i.e., DHA, DPA3 and EPA. Its formulation as a lipid microemulsion delivery system may well be the key factor contributing to its high absorption. Studies into the absorption of n-3 FAs in an emulsified state found that delivering n-3 FAs in this form increased the surface area of the oil, allowing for increased access for digestion lipases, thus leading to improved absorption. Those same studies showed that the AUC_0-24_ of total EPA and DHA was 6.1 times higher in the lipid emulsion formulation than in the regular unformulated oil [58].

A strength of the current study is the comprehensive investigation into both the PKs of n-3 FAs as well as their metabolites. However, as previously mentioned, one limitation of this study is that a longer observation period should have been applied due to the higher bioavailability of LMF, which still showed significantly higher n-3 FA blood concentrations at 24 h compared to the other, non-micellar, treatments. Considering the delayed metabolism of LMF, future studies may incorporate this factor by extending the observation period or adjusting the dosing regimen accordingly.

## Figures and Tables

**Figure 1 metabolites-14-00265-f001:**
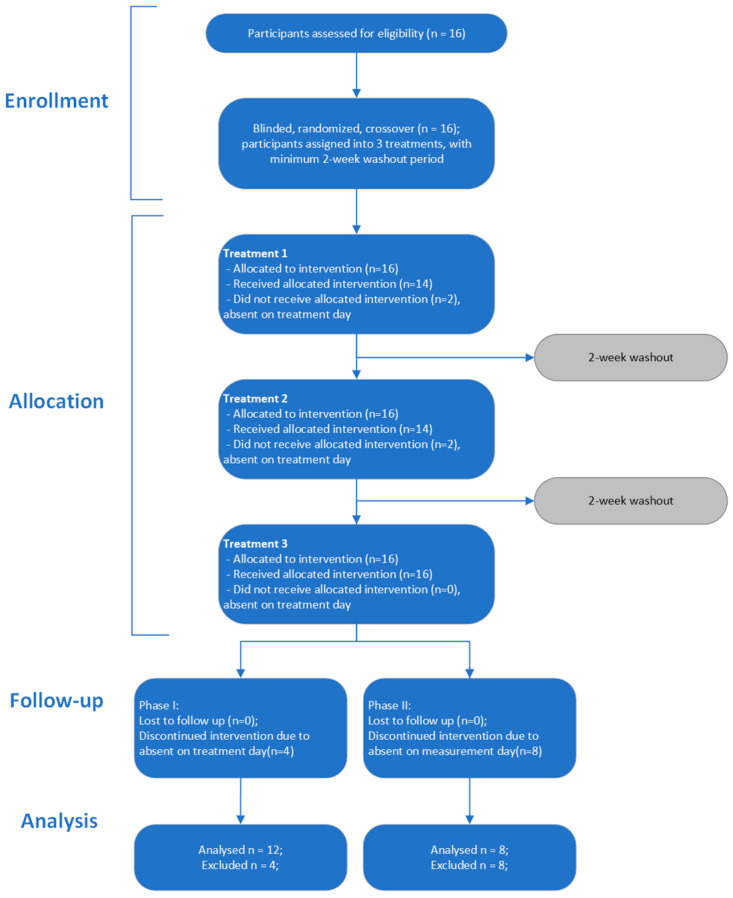
Study process flowchart.

**Figure 2 metabolites-14-00265-f002:**
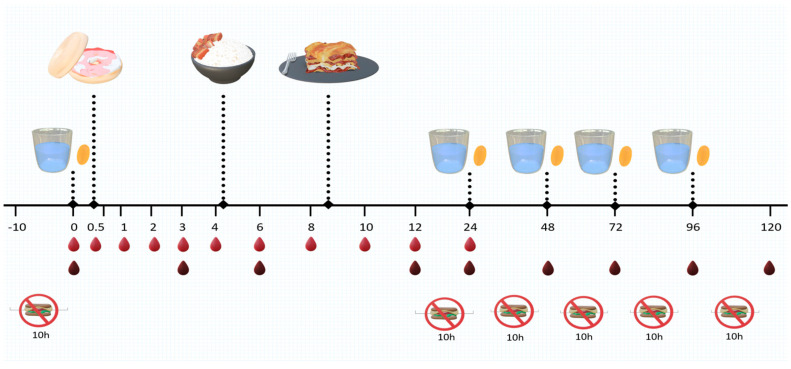
Treatment timeline. For bioavailability, blood samples were collected at 0, 0.5, 1, 2, 3, 4, 6, 8, 12, and 24 h (bright red). Breakfast was provided after the participants consumed the capsules, lunch after the 4 h blood sample, and dinner after the 8 h blood sample. For blood lipids, samples were collected at 0, 3, 6, 12, 24, 48, 72, 96, and 120 h (dark red) after the initial dose. Participants carried out an overnight fast prior to the 0, 24, 48, 72, 96, and 120 h collection, and, except for 120 h, were given the intervention immediately after blood sample collection.

**Figure 3 metabolites-14-00265-f003:**
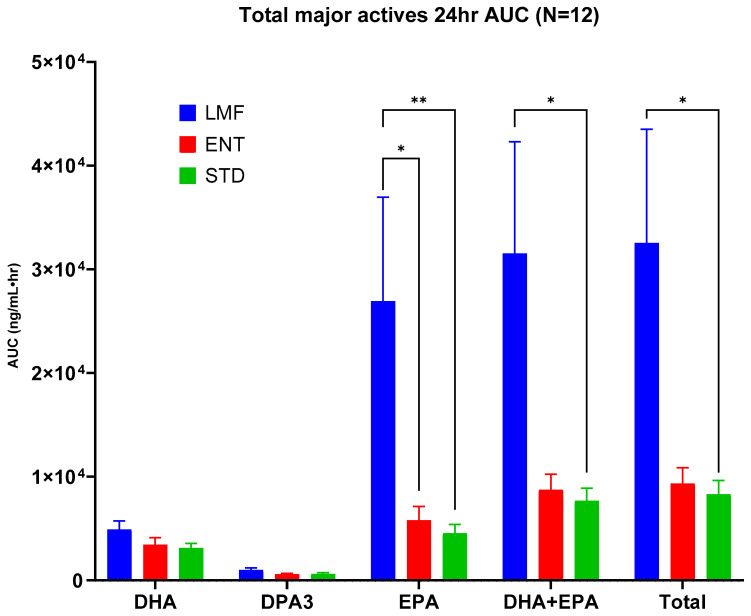
Mean blood concentration (AUC_0-24_) of EPA, DPA3, and DHA after treatment with each Omega-3 formulation; * *p* ≤ 0.05, ** *p* ≤ 0.01, repeated measure two-way ANOVA and Tukey’s multiple comparison test; *n* = 12.

**Figure 4 metabolites-14-00265-f004:**
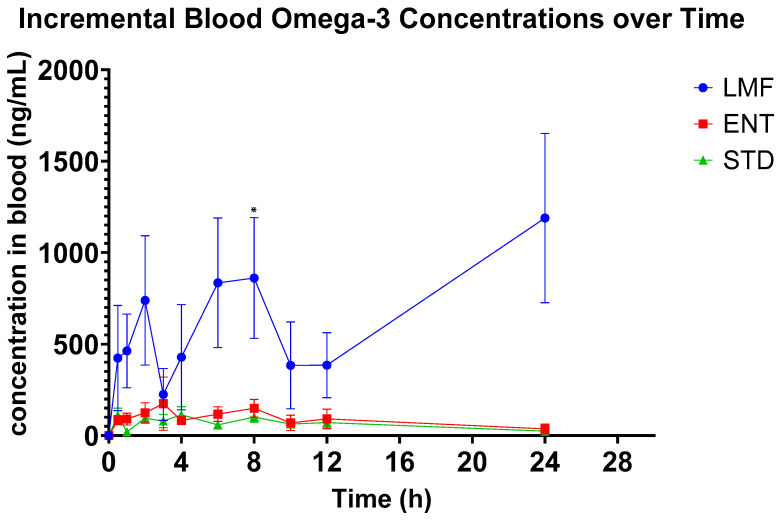
Average incremental concentrations (iAUC24) of total Omega-3 fatty acids in participants’ blood after oral ingestion of assigned intervention; *n* = 12; * denotes *p* ≤ 0.05, repeated measure two-way ANOVA followed by Tukey’s multiple comparison test.

**Figure 5 metabolites-14-00265-f005:**
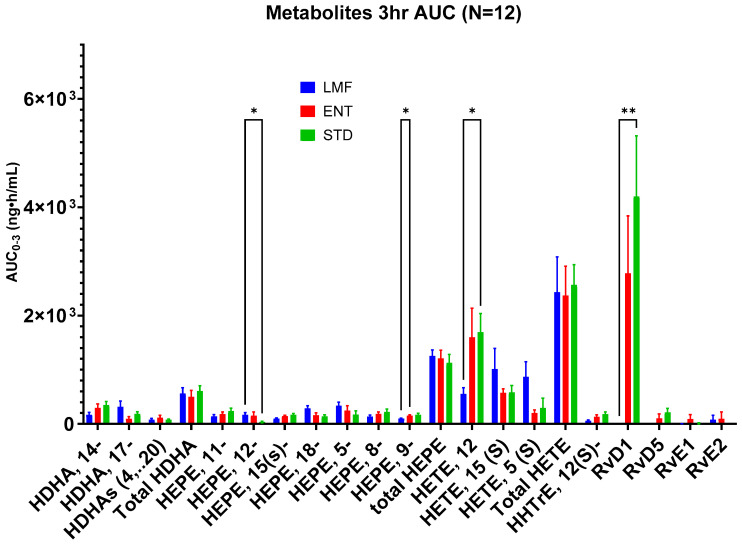
Mean total blood concentrations of metabolites 3 h after administration of each Omega-3 formulation; * denotes *p* ≤ 0.05, ** *p* ≤ 0.01, repeated measure two-way ANOVA with Tukey’s multiple comparison test.

**Figure 6 metabolites-14-00265-f006:**
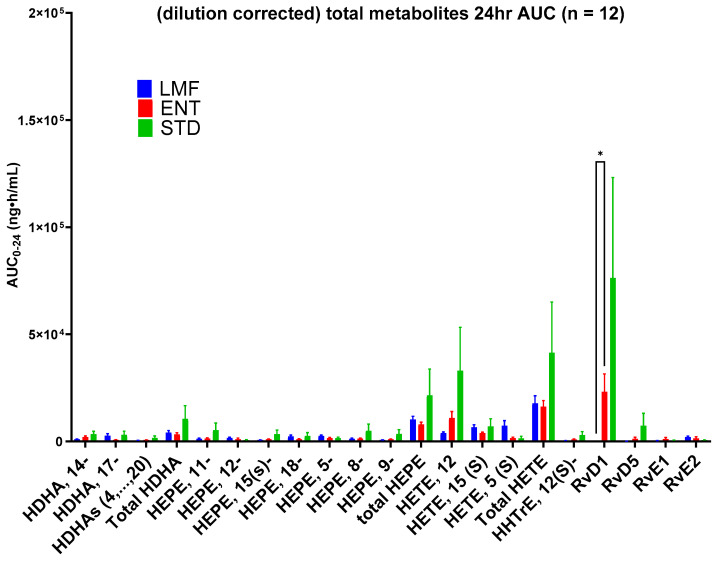
Mean total blood concentrations of metabolites 24 h after administration of each Omega-3 formulation; * denotes *p* ≤ 0.05, repeated measure two-way ANOVA with Tukey’s multiple comparison test.

**Figure 7 metabolites-14-00265-f007:**
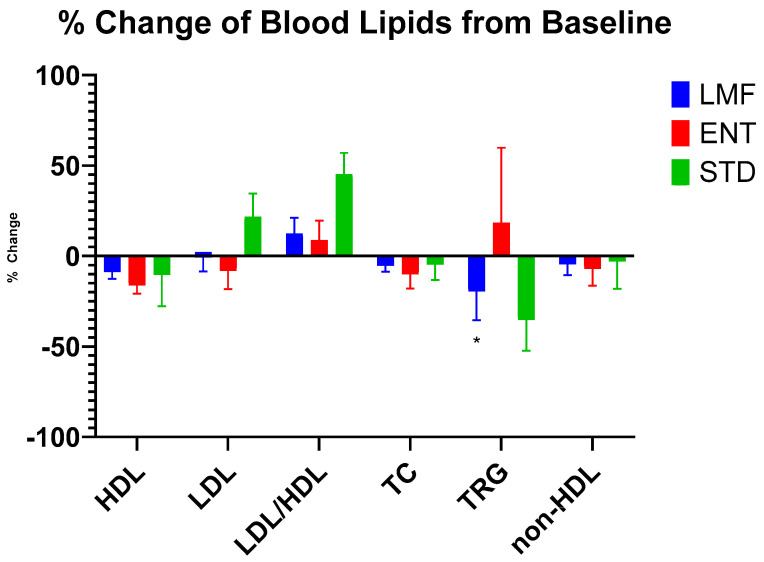
Changes in blood lipids over 120 h following the daily administration of individual n-3 fish oil treatments; * denotes *p* ≤ 0.01 with repeated measure two-way ANOVA performed on individual data using Time 0 as Control with Dunnett’s multiple comparison correction; *n* = 8.

**Table 1 metabolites-14-00265-t001:** Participant eligibility criteria.

Inclusion Criteria	Exclusion Criteria
≥18 years old	Less than 18 years old
Good physical condition	Smokers
Provide signed informed consent	Taking prescribed medication
	Liver disease
	Kidney disease
	Gastrointestinal disease
	Allergy to berberine
	Intend to become pregnant, pregnant, or breast-feeding

**Table 2 metabolites-14-00265-t002:** Standardized meal plan.

Day	Breakfast	Dinner
Monday	Bagel with cream cheese and jam	Lasagna and salad
Tuesday	Bagel with cream cheese and jam	Pork chops, sweet potatoes, beans, salad
Wednesday	Bagel with cream cheese and jam	Chicken skewers, pita bread, tzatziki, cucumbers, tomatoes
Thursday	Bagel with cream cheese and jam	Lamb chops, cauliflower, green beans
Friday	Bagel with cream cheese and jam	Beef fajita, onions, peppers, wheat tortilla wrap

**Table 3 metabolites-14-00265-t003:** Chemical standards and scan parameters for Q exactive Orbitrap mass spectrometer.

Compounds	Chemical Formula [M]	Scanned Mass [*m*/*z*]	Start [min]	End [min]
18-HEPE	C20H30O3	317.21222	4.00	4.60
15-HETE-d8	C20H24D8O3	327.27808	4.50	5.10
17-HDHA/14-HDHA	C22H32O3	343.22787	7.00	8.30
DPA3	C22H34O2	329.2486	9.10	9.90
EPA	C20H30O2	301.2173	8.60	9.20
DHA	C22H32O2	327.23295	9.00	9.60
(18S-) Resolvin E1	C20H30O5	349.20205	3.00	4.70
(18S-) Resolvin E2	C20H30O4	333.20713	4.30	6.50
(AT-)Resolvin D2,3,4	C22H32O5	375.2177	4.70	7.70
Resolvin D5, 6, Protectin D1, Maresin	C22H32O4	359.22278	6.00	8.20
EPA-d5	C20H25D5O2	306.24869	6.80	7.25
15(S), 12(S), 5(S)-HETE	C20H32O3	319.22787	7.10	8.10
12(S)-HHTrE	C17H28O3	279.19657	6.30	6.90
Prostaglandin D2, E2	C20H32O5	351.2177	3.80	5.20
Protaglandin F2a	C20H34O5	353.23335	4.70	5.30
Thromboxane B2, 6-keto protaglandin F1a	C20H34O6	369.22826	3.50	5.50
DHA-d5	C22H27D5O2	332.26434	9.00	9.60
EPA Oxylipins	C20H30O3	317.21222	6.70	7.70

**Table 4 metabolites-14-00265-t004:** Study participants’ baseline characteristics, presented in average ± SEM.

Parameter	
*N*	16
Males | Females	9 | 7
Age (years)	36.1 ± 2.4
Weight (kg)	61.8 ± 2.4
BMI (kg/m^2^)	22.1 ± 0.5
DHA (ng/mL)	1007.1 ± 131.1
EPA (ng/mL)	3085.2 ± 833.4
DPA (ng/mL)	215.0 ± 30.5

**Table 5 metabolites-14-00265-t005:** Blood concentrations over 24 h.

AUC_0-24_ ng/mL·h	STD	ENT	LMF
DHA	3136.6 ± 433.3	3436.8 ± 688.0	4910 ± 827.5
DPA3	626.4 ± 120.3	590.6 ± 93.4	1016.8 ± 181.8
EPA	4545.4 ** ± 839	5798 * ± 1317	26,920 ** ± 10,021
DHA + EPA	7682 * ± 1225	8734 ± 1503	31,529 * ± 10,783
Total	8309 * ± 1330	9325 ± 1555	32,546 * ± 10,950

Means ± SEM reported; * *p* ≤ 0.05, ** *p* ≤ 0.01, repeated measure two-way ANOVA and Tukey’s multiple comparison test; *n* = 12.

**Table 6 metabolites-14-00265-t006:** Incremental blood concentration over 24 h of total n-3 FAs.

	STD	ENT	LMF
iAUC (ng/mL·h)	1498.9 **** ± 443.0	2057.2 **** ± 813.7	16,150 **** ± 5454
C_max_ (ng/mL)	186.0 ± 44.8	389 ± 141	1732.8 ± 478.3
T_max_ (hr)	3.9 ± 1.1	4.7 ± 1.2	10.0 ± 2.1

Means ± SEM reported; **** *p* ≤ 0.0001, repeated measure ANOVA with Tukey’s multiple comparison test; *n* = 12.

**Table 7 metabolites-14-00265-t007:** Blood concentrations (dilution corrected) of metabolites over 3 h.

AUC_0-3 h_ ng·h/mL	STD	ENT	LMF
HDHA, 14-	347.2 ± 68.5	292.0 ± 75.6	167.1 ± 47.7
HDHA, 17-	182.9 ± 43.6	93.3 ± 41.0	316.0 ± 106.2
HDHAs (4…20)	73.3 ± 15.9	113.7 ± 43.8	76.2 ± 26.2
Total HDHA	603.5 ± 101.5	498.9 ± 120.0	559.3 ± 109.4
HEPE, 11-	234.9 ± 54.7	180.2 ± 39.3	140.3 ± 30.0
HEPE, 12-	33.6 * ± 14.1	150.6 ± 73.0	166.3 * ± 40.7
HEPE, 15-	165.3 ± 27.4	141.2 ± 16.2	95.3 ± 14.3
HEPE, 18-	135.4 ± 31.3	157.6 ± 49.0	285.6 ± 51.2
HEPE, 5-	169.3 ± 70.8	246.5 ± 86.9	335.9 ± 66.1
HEPE, 8-	220.7 ± 53.2	182.8 ± 37.0	134.6 ± 27.9
HEPE, 9-	165.7 * ± 30.0	149.8 * ± 17.3	95.8 ± 11.2
total HEPE	1124.9 ± 158.2	1208.8 ± 152.8	1253.9 ± 109.2
HETE, 12-	1692.9 * ± 342.5	1597.2 ± 539.4	551.1 * ± 118.5
HETE, 15-	579.2 ± 129.3	572.7 ± 75.1	1011.7 ± 381.0
HETE, 5-	293.0 ± 181.5	201.5 ± 55.6	871.6 ± 274.5
Total HETE	2565.1 ± 373.7	2371.4 ± 538.3	2434.3 ± 650.3
HHTrE, 12-	179.4 ± 43.3	131.2 ± 35.4	53.9 ± 19.2
RvD1	4197 ** ± 112	2782 ± 1058	0.2 ** ± 0.2
RvD5	212.3 ± 74.9	100.6 ± 83.7	−23.1 ± 23.1
RvE1	−2.7 ± 22.2	89.3 ± 80.5	−16.5 ± 19.9
RvE2	−91.8 ± 65.7	94.8 ± 125.8	77.8 ± 80.5

Means ± SEM reported; * denotes *p* ≤ 0.05, ** *p* ≤ 0.01, repeated measure two-way ANOVA with Tukey’s multiple comparison test; *n* = 12.

**Table 8 metabolites-14-00265-t008:** Blood concentrations (dilution corrected) of metabolites over 24 h.

AUC_0-24 h_ ng·h/mL	STD	ENT	LMF
HDHA, 14-	3410 ± 1296	2041.0 ± 493.8	998.7 ± 172.2
HDHA, 17-	3033 ± 1759	621.7 ± 228.1	2637.3 ± 1031.2
HDHAs (4…20)	1618.5 ± 951.7	630.8 ± 122.9	439.6 ± 112.4
Total HDHA	10,450.5 ± 6273.9	3293.4 ± 752.5	4075.5 ± 1026.2
HEPE, 11-	5224.1 ± 3330.8	1292.5 ± 329.0	1192.1 ± 318.6
HEPE, 12-	562.9 ± 293.0	1016.5 ± 552.0	1641.6 ± 371.8
HEPE, 15-	3345.0 ± 1960.2	970.1 ± 144.1	636.9 ± 77.6
HEPE, 18-	2527.3 ± 1561.8	947.8 ± 283.6	2317.7 ± 631.3
HEPE, 5-	1465.6 ± 533.4	1447.9 ± 448.4	2533.6 ± 436.7
HEPE, 8-	4931.9 ± 3164.2	1288.3 ± 310.1	1226.0 ± 298.5
HEPE, 9-	3438.7 ± 2086.5	960.1 ± 161.0	629.9 ± 64.2
total HEPE	21,495.5 ± 12,252.5	7923.2 ± 1028.3	10,177.8 ± 1522.0
HETE, 12-	32,927.3 ± 20,323.3	10,891.8 ± 3080.1	3905.1 ± 534.2
HETE, 15-	6983.2 ± 3602.3	3846.1 ± 453.5	6596.8 ± 1193.3
HETE, 5-	1446.9 ± 966.5	1510.8 ± 488.4	7313.2 ± 2351.5
Total HETE	41,357.5 ± 23,732.8	16,248.7 ± 2824.7	17,815.2 ± 3486.3
HHTrE, 12-	2899.3 ± 1669.9	867.6 ± 244.8	361.7 ± 58.3
RvD1	91,591 * ± 55,336	30,786 ± 9988	0.19 * ± 0.17
RvD5	7336 ± 5788	2050 ± 1245	39.9 ± 107.7
RvE1	662 ± 313	1628 ± 1130	211.6 ± 201.2
RvE2	785.8 ± 341.5	2097 ± 1135	1854 ± 827

Means ± SEM reported; * denotes *p* ≤ 0.05, repeated measure two-way ANOVA with Tukey’s multiple comparison test; *n* = 12.

## Data Availability

The original contributions presented in the study are included in the article, further inquiries can be directed to the corresponding authors.

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
