# Peer review of "Evaluation of the Metabolite Profile of Fish Oil Omega-3 Fatty Acids (n-3 FAs) in Micellar and Enteric-Coated Forms—A Randomized, Cross-Over Human Study"

_metabolites, 2024, doi:10.3390/metabo14050265_

Round 1

Reviewer 1 Report

Comments and Suggestions for Authors

This MS is deserving publication. I would like note that in this MS compared too different formulations, which are differ in many aspects. First, formulations - standard soft gels (STD) 1000 mg fish oil (FO) providing 600 mg n-3 FA; 2 enteric coated soft gels (ENT) 1170 mg FO providing 600 mg n-3 FA; 3) a new micellar formulation (LMF) 585 mg FO providing 374 mg n-3 FA. Second is amount of n-3 fatty acids – 600, 600 and 374 mg for STD, ENT and LMF, respectively. Third, ratio EPA/DHA is 2:1 for STD and ENT, and 2:1.5 for LMF. These formulations contain significant doses of different fish oils, which have numerous fatty and potentially influence on experimental data.

Authors understand some shortages of this investigation.

Some notes to MS 2997737

Abstract

Abbreviations like “AUC” better explain at first appearance.

There are two isomers of docosapentaenoic acid 22:5(n-3) and 22:5(n-6). Respectively abbreviations are DPA3 and DPA6.

Line 34 – formally for human essential fatty acids are cis-linoleic acid (LA) and alpha-linolenic    acid (ALA). Probably better use term “essential” for them.

Line 77 – what means “the larger fats”? Is it triacylglycerol droplets?

Materials Methods

            As for formulations used different fish oils (anchovy, tuna, sardine, and/or mackerel) with different fatty acid compositions, it looks better give full data on oil FA compositions.

Why for STD and ENT no data on DPA3 acid? In Fig. 3 and Table 5. shown DPA concentrations for all formulations.

Thermo Q exactive Orbitrap is more correct than “Thermo Qexactive Orbitrap”

Results

Line 297. What means non-HDL cholesterol? Is it free cholesterol?

In Supplement are shown only chromatograms of standards. It is interesting to look on chromatogram of real blood sample.

Reviewer 2 Report

Comments and Suggestions for Authors

The main focus of this manuscript is the description of whole blood omega-3 fatty acid and oxylipin concentrations following single dosing of one of three formulations of omega-3 supplements. These are standard, enterically coated, and a formula designed to promote emulsification and absorption. The study was conducted in a small number of healthy participants using a cross-over design.  The study is well designed and well conducted but there is quite high loss of participants. The novel formulation lead to greatly increased EPA appearance in the bloodstream, but surprisingly this was not seen for DHA. 

Specific comments:

1. Line 56, 104 and elsewhere. HETEs are NOT formed from EPA; they are formed from arachidonic acid. This needs correcting throughout.

2. Line 58 and 63. There are multiple HETEs; so this should be written as plural or the specific HETEs named.

3. Line 77. Language is way too simplistic. Bile acids do not "break down" fats; they aid emulsification. What are "larger fats"? Do you mean lipids with esterified fatty acids? What are smaller molecules (Fatty acids?)?

4. Line 78. "small intestinal lining"? Please be scientific.

5.  Line 83-84. You says "delayed release forms" have not been studied, but they have. You mention this and cite studies about it on lines 90-96.

6. Line 106-108. This statement is absolutely not true. Both chemical and physical formulations have been studied before; indeed you cite a number of studies on this.

7. Line 116. Please provide a statement on the chemical form of omega-3s in each supplement.

8. Line 128. Delete "with"

9. Line 134-135. Delete "(a more bioavailable formulation)" because that is what you are testing so you could not have known that when planning the study.  

10. Line 157. Please state the delay between consuming the capsule and eating the bagel.

11. Section 2.3. Please state what is being collected by SPE. Is it fatty acids? and only fatty acids?

12. Line 231-233. Please provide the reasonsLMF. The later paeak for drop-out.

13. Figure 4. I see these findings a little differently from the authors who emphasise a late peak. In fact there is an early peak (about 2 hours) suggesting more rapid absorption of EPA from the LMF. The later peak (at 6 and 8 hours) follows from the lunch at 4 hours and suggests release of EPA from intestinal epithelial cells where it has been trapped. This phenomenon has been described previously.  The later appearance (24 hours) suggests preferential release of newly acquired EPA from the liver back into the bloodstream. Again this phenomenon has been previously described.

14. Figure 5. Do you think these metabolites are in the supplements?

15. Table 7 and 8. HDHA's -> HDHAs (it is plural not possessive)

16. Fig 7. Why is the effect of STD on TGs not significant? The effect size is bigger than for LMF and the error is similar.

17. Remove Table 9 and just say there were no AEs.

18. Line 359-360. It is unclear where this figure of 2 g/day comes from. The reviewer has searched the ODS site and does not see it. Indeed the ODS statement on omega-3s says "according to the European Food Safety Authority, long-term consumption of EPA and DHA supplements at combined doses of up to about 5 g/day appears to be safe [187]. It noted that these doses have not been shown to cause bleeding problems or affect immune function, glucose homeostasis, or lipid peroxidation. Similarly, FDA has concluded that dietary supplements providing no more than 5 g/day EPA and DHA are safe when used as recommended [188].". Please correct your statement.

19. Line 369. what do you mean by "delayed elimination from the body".

20. Line 439. Where did ISURA get the funds from? There is a need to be more transparent.   

Round 2

Reviewer 2 Report

Comments and Suggestions for Authors

A very good revision that addresses the comments raised. Authors still do not say who funded the trial.

Author Response

Comments and Suggestions for Authors

A very good revision that addresses the comments raised. Authors still do not say who funded the trial.

Dear Reviewer, thank you very much for your feedback!

No external funding was provided for this study. Open Access was provided by ISURA's research fund as part of its non-profit mandate (as mentioned, ISURA generates revenue through its contaminant testing services). However, we do acknowledge that The Factors Group of Nutritional Companies supplied the LipoMicel™ and Enteric Softgel samples for research and testing. -

Please see the revised “Funding” statement in the paper.